# Study on the Printability through Digital Light Processing Technique of Ionic Liquids for CO_2_ Capture

**DOI:** 10.3390/polym11121932

**Published:** 2019-11-23

**Authors:** Matteo Gillono, Annalisa Chiappone, Lorenzo Mendola, Manuel Gomez Gomez, Luciano Scaltrito, Candido Fabrizio Pirri, Ignazio Roppolo

**Affiliations:** 1Department of Applied Science and Technology, Politecnico di Torino, Corso Duca degli Abruzzi 24, 10129 Torino, Italy; matteo.gillono@iit.it (M.G.); annalisa.chiappone@polito.it (A.C.); lollomen92@live.it (L.M.); manuel.gomezgomez@polito.it (M.G.G.); luciano.scaltrito@polito.it (L.S.); fabrizio.pirri@polito.it (C.F.P.); 2Center for Sustainable Future Technologies @Polito, Istituto Italiano di Tecnologia, Via Livorno 60, 10144 Torino, Italy

**Keywords:** ionic liquids, 3D printing, digital light processing, CO_2_ capture

## Abstract

Here we present new 3D printable materials based on the introduction of different commercially available ionic liquids (ILs) in the starting formulations. We evaluate the influence of these additives on the printability of such formulations through light-induced 3D printing (digital light processing—DLP), investigating as well the effect of ionic liquids with polymerizable groups. The physical chemical properties of such materials are compared, focusing on the permeability towards CO_2_ of the different ILs present in the formulations. At last, we show the possibility of 3D printing high complexity structures, which could be the base of new high complexity filters for a more efficient CO_2_ capture.

## 1. Introduction

Public and political opinion is currently always more interested in global warming phenomenon. Data collected through the last 150 years demonstrate that the greenhouse effect is the main reason of the global temperature increase and carbon dioxide is one of the main gases involved in this effect. The emission of a huge quantity of this gas (nearly 40 Gtonn/year in 2018 only from fossil fuels) [1] is causing an abrupt increase of its concentration from 300 ppm in the 50s to more than 400 ppm currently [2,3]. So, the capture of the gases present in the atmosphere is an urgent demand in order to decrease the greenhouse gases emissions, in particular for carbon dioxide. In this context, different technologies are available for CO_2_ capture (CC), such as pre-combustion, post-combustion and oxyfuel capture [4,5,6,7,8]. These technologies can employ liquids, both in pre-combustion and post-combustion processes [9,10], but also solid materials in form of beads and powders packed in columns, used as gas contactors [11]. Such materials can undergo two main capture processes: physical adsorption and chemical absorption. In the physical adsorption process, the capture is mainly due to Van der Waals forces generated by the chemical potential and the affinity between the CO_2_ molecules and the surface of the sorbents. The weak bond created guarantees a minimum amount of energy to be used in the regeneration phase. The most used solid CO_2_ adsorbents are zeolites, activated carbons and metal organic framework (MOF) [12,13,14,15]. The chemical absorption process is more employed in post combustion processes, and involves the creation of strong irreversible chemical bonds that bring to the creation of new chemical species. The main problem related to this process is the high amount of energy required to break the chemical bonds to regenerate the CO_2_ absorbed molecules. Among the most used chemical absorbents, there are the amine and lithium based materials [16,17,18,19].

In the last years, ionic liquids (ILs) have gained a lot of attention in the application of CC technologies [5,20,21,22]. They are salts with low melting point, usually under 100 °C and below room temperature for RTILs (room temperature ionic liquids); they possess very low vapor pressure with a consequent almost zero volatility. Furthermore, their cations and anions can be tuned and engineered in such a way that their CO_2_ solubility can be increased. ILs have been largely used in adsorbing liquid blends, like aqueous alkanolamines. The main differences of these systems consist in a decreased degradation in oxidizing environments, minor corrosion rate and lower regeneration energy [23]; nevertheless, the ILs present problems caused by their high viscosity and lower CO_2_ solubility compared with standard capturing liquid systems [24,25]. Supported liquid membranes (SLM), solid membranes containing ILs, have been also proposed [26,27,28,29]. Among those, some are formed of ionic liquids that were functionalized with reactive groups and used to create polymeric chains, the so-called poly ionic liquids (PILs) [30,31,32,33]. In 2007 Noble et al. proposed for the first time the use of PILs based membranes for CO_2_ capture, obtaining better results than the ones of the relative neat ILs [34]. Merging the intriguing CO_2_ solubility ability of ILs with the handiness of solid materials results extremely attractive for the development of CO_2_ capturing devices [35]; furthermore, the possibility to shape the material with complex geometries could enable the production of smart filters. In this perspective 3D printing can be a forwarding solution [36].

3D printing is a fabrication method that enables the direct fabrication of three dimensional objects starting from a digital model [37]. Compared to traditional methods, 3D printing enables a higher level of complexity, gathered with saving of raw materials and lower energy consumption [38]. ILs were already used in 3D printing both for direct Ink Writing (DIW) [39,40,41], Fused Filament Fabrication (FFF) technology [42] and in photopolymerization-based technologies (i.e., stereolithography—SL—and digital light processing—DLP) [43]. The union between ILs and 3D printing was explored in many applications, such as sensors [39,44], optical data storage systems [45], chemical [46] and biomedical devices [47].

Among the different 3D printing technologies, DLP results particularly interesting for the production of functional components, in which materials and design properties are gathered. DLP is a vat polymerization 3D printing method that employs photosensitive liquid resins able to be cure upon light irradiation. In a DLP equipment, a digital micromirror device (DMD) shines the light on a photocurable formulation following a two-dimensional pixel pattern, which represents a single layer of a digital model (CAD file), inducing the layer solidification. Repeating the same procedure along the z axis, the object is built [48]. If compared to the other 3D printing technologies, DLP presents one of the highest resolution together with high output [49]. The polymeric materials typically employed in DLP are acrylic- and methacrylic- resins, which are cured by means of free-radical photopolymerization [50]. Considering the wide range of suitable resins, objects produced by DLP can now cover a wide range of applications, from biomedical to mechanical, electronic, energy and many more [51].

In this paper, we present the development of polymeric formulations printable with the DLP technique containing commercial available ILs. Those ILs are particularly suitable for DLP technology due to their intrinsic characteristic: first, in DLP liquid photocurable formulations are processed, so perfectly compatible with RTILs; second, cross-linked polymers are produced by this technology, trapping ILs and possibly avoiding phase separation or leakage; third, DLP allows a higher shape freedom and superior precision, which can be straightforward in order to produce complex devices containing ILs [52]. Here, we selected six commercially available ILs (Bmim[BF_4_], Bmim[Tf_2_N], Bvim[Tf_2_N], Amim[Tf_2_N], Bmim[ac] and C_4_N_MA,11_[Tf_2_N], Table 1) and we fabricated 3D printed structures based on these compounds, both trapped and co-polymerized in the polymeric network. First, a polymerization study showing the photo-rheological and infrared spectroscopy kinetics is presented to assess the reactivity of the formulations containing different ILs. Second, the formulations are loaded in the DLP system and 3D printed structures. The thermo-mechanical properties of the printed structures were then investigated. Afterwards, CO_2_ permeability on photo-polymerized membranes is done to study their gas transport properties. At last, very complex geometries (gyroids) are 3D printed.

## 2. Materials and Methods

### 2.1. Materials

Poly(ethyleneglycol)diacrylate (PEGDA, Mn. 575), (Bis(2,4,6-trimethylbenzoyl)-phenylphosphineoxide (BAPO) and Reactive Orange 16 were purchased from Merck (Milan, Italy) and used as received. Regarding the ionic liquids employed, 1-Butyl-3-Methylimidazolium tetrafluoroborate, 1-Butyl-3-Methylimidazolium bis (trifluorometilsulfonyl)imide, 1-Allyl-3-Methylimidazolium bis (trifluoromethylsulfonyl)imide and 1-Butyl-3-Methylimidazolium acetate were purchased from Merck (Milan, Italy); *N*,*N*,*N*,*N*-ButyldimethylMethacryloyloxyethylammonium bis(trifluoromethylsulfonyl)imide and 1,4-Butandiyl-3,3′-bis-1-vinylimidazoliumbis (trifluoromethylsulfonyl)imide were purchased from Solvionic (Toulouse, France). The ionic liquids used are summarized in Table 1.

### 2.2. Formulation Preparation and Printing

A reference formulation (named PEGDA) was obtained by mixing Poly(ethyleneglycol)diacrylate with Reactive Orange 16 at 0.2 phr (per hundred resin), as a dye, and BAPO at 2 phr as a photoinitiator.

The formulations containing the different ILs were produced by adding 0.4 mmol/g of each IL to the reference formulation. These were named as P_Bmim[BF_4_], P_Bmim[Tf_2_N], P_Bmim[ac], P_C_4_N_MA,11_[Tf_2_N], P_Amim[Tf_2_N] and P_Bvim[Tf_2_N] accordingly to the IL used. The influence of the IL concentration was tested for Bmim[BF_4_] and for C_4_N_MA,11_[Tf_2_N], in this case multiples of 0.4 mmol/g were added to the reference formulation. The samples were named as P_(ILname)_X, where X is the multiple of 0.4 used.

The different formulations were then printed using a DLP printer HD 2.0 (Robot Factory S.r.l., Venice, Itay), equipped with a projector as light source (X–Y resolution 50 µm corresponding to 1920 pixels × 1080 pixels, intensity 10 mW/cm^2^). The layer thickness was set to 50 µm and the layer exposure time was set for each sample according to preliminary tests (values reported in Table 2). The obtained samples were first washed, gently rinsing in ethanol, and then underwent a post curing process (5 min), performed with a medium pressure mercury lamp also provided by Robotfactory equipped with a rotating platform (intensity 10 mW/cm^2^).

### 2.3. Characterization

FTIR-ATR spectra were collected using a Tensor 27 FTIR Spectrometer (Bruker, Billerica, MA, USA) equipped with ATR tool, 32 scans were collected with a resolution of 4 cm^−1^ from 4000 to 400 cm^−1^. Real-time photo-rheology measurements were performed using an Anton Paar rheometer Physica MCR 302 (Graz, Austria) in parallel plate mode equipped with a bottom glass plate. For the measurement a light source Hamamatsu Photonic LC8 lamp (Hamamatsu City, Japan) with visible bulb and a cut-off filter below 400 nm equipped with 8 mm light guide) was placed below the glass plate. The upper aluminum plate was placed at 0.1 mm from the glass one and the sample was kept at a constant temperature (25 °C) and under constant shear frequency of 5 rad s^−1^ at 1% of amplitude accordingly to preliminary amplitude sweep tests; light was turned on after 1 min in order to stabilize the system. Dynamic mechanical thermal analyses (DMTAs) were performed with a Triton Technology TTDMA (Mansfield, MA, USA). All of the experiments were conducted with a temperature ramp of 3 °C/min, applying a force with a frequency of 1 Hz and a displacement of 20 μm with a temperature range between −80 and 30 °C. TGA analysis were carried out with a Themo Netzsch TG 209 F1 Libra (Selb, Germany) instrument with a temperature ramp of 10 °C/min in air. For the permeability measurements, a single chamber Extrasolution Multiperm permeometer was used (Extrasolution, Pieve Fosciana, Italy). For the CO_2_ uptake measurements, FT-IR was used to check the interaction between the samples and the gas. The apparatus for CO_2_ atmosphere is composed by a pressurizing test chamber connected with a CO_2_ cylinder and a vacuum pump used to clean the chamber and regenerate the samples. A thin layer of the printable formulations (25 μm) was deposited on the silicon wafer and photocured in nitrogen atmosphere. The samples were then degassed in vacuum (5 mbar) for 24 h to desorb the gases adsorbed. The CO_2_ uptake phase was carried out by inserting the samples in the CO_2_ unit for 4 h at a pressure of 4 bar. Afterwards, the samples were again subjected to a vacuum cycle (24 h at 5 mbar) and a further heating (80 °C). After every step, the samples were tested by FT-IR in order to check the CO_2_ peaks in the 2400–2200 cm^−1^ range.

## 3. Results and Discussion

In order to determine the printability of the different formulations, investigations on the polymerization of the prepared mixtures containing the different ionic liquids were performed. The photo-rheological measurements show how the liquid formulations behave during light irradiation; in particular, the variation of the elastic modulus G′ vs. the irradiation time was investigated. The tests, reported in Figure 1, showed that three of the chosen ILs (i.e., Bmim[BF_4_], Bmim[Tf_2_N] and Bvim[Tf_2_N]) do not influence the kinetic of polymerization of the monomer, indicating that neither the imidazole cation nor the corresponding anions inhibit the polymerization process. Instead, C_4_N_MA,11_[Tf_2_N] showed slightly slower kinetics, which led to a delayed plateau value of G′, probably due to the ammonium-based cation, which could induce radical quenching [53]. Moreover, the IL Amim[Tf_2_N], which bears an allyl group, slows the polymerization: this could be expected considering the poor reactivity of allyl monomers [54,55]. At last, it was not possible to measure the polymerization of P_Bmim[ac]: in this case no variation of the G’ modulus was observed, indicating that the acetate-based IL completely hinders the reaction. The data obtained gave an indication of the required layer exposure time during printing that was then experimentally adjusted (Table 2).

After this preliminary study, 3D printing tests were performed to create both planar films and three-dimensional filter-like structures with cubical structures (Figure 2). The cubical hollow structure was chosen both to prove the possibility to manufacture complex features with high resolution and to create a structure with a high surface area to improve the gas contact with the material during the CO_2_ uptake analysis.

The key parameters to control in the printing process are the layer thickness and the layer exposure time. In fact, to adjust both parameters is essential to obtain the best printing performance, resulting in increased-X, Y and Z resolution. The exposure times (Table 2) were empirically obtained during the production of membranes and 3D printed parts, considering a good trade-off between ease of handling and precision of printing (i.e., no over-polymerization); those values are in good agreement with the trends obtained from photorheology tests, showing that all the samples containing IL required layer exposure times similar to neat PEGDA monomer. The only exception was the sample P_Amim[Tf_2_N], which needed longer exposure times for the polymerization in the form of membrane, and it demonstrated unsuitability to obtain more complex shapes. Sample Bmim[ac] could not be polymerized thus it was not further considered for the next experiments.

FTIR-ATR analyses of the 3D printed ionic liquids blended with PEGDA were carried out in order to further investigate the interaction of the different ILs with PEGDA matrix, in particular to observe weather if the ILs containing double bonds (C_4_N_MA,11_[Tf_2_N], Bvim[Tf_2_N] and Amim[Tf_2_N]) co-polymerize. First, the FTIR-ATR spectra of all neat ionic liquids and PEGDA as reference were collected (Appendix A), then the polymerized samples of each formulation were analyzed before and after being washed in acetone for 24 h (Appendix A). The spectra confirmed that C_4_N_MA,11_[Tf_2_N] and Bvim[Tf_2_N] copolymerized with PEGDA acrylic bonds, proving that these ionic liquids remained linked to the polymer chains during the polymerization reaction, being a part of polymeric network. So, the structures with these ILs could be used even in liquid environment without leakage of the components. Instead, in the spectra of the samples containing the remaining ILs (Bmim[BF_4_], Bmim[Tf_2_N] and Amim[Tf_2_N]), we observed that the peaks related to the ionic liquids disappeared after solvent rinsing, demonstrating that they were only trapped in the polymer matrix and no reaction occurred during polymerization. This confirms also that the allyl group in Amim[Tf_2_N] was not reactive enough to react with photoactivated process. Moreover, the weight of the samples was measured before and after the solvent washing. Table 3 reports the values of the concentration of the ionic liquids in the samples and the loss in weight after solvent washing. The results confirmed the considerations reported for FT-IR measurements.

The evaluation of the effect of the presence of the ILs on the thermo-mechanical properties of the 3D printed structures was carried out by DSC and DMTA. The data are reported in Table 4 while the curves are reported in Appendix A in the supporting information file. Starting from neat PEGDA as reference materials, it was possible to distinguish two distinct behavior: while the ILs that do not react with PEGDA (Bmim[BF_4_], Bmim[Tf_2_N] and Amim[Tf_2_N]) induced a decrease of the *T*_g_, for the sample in samples in which copolymerization occurred, an increase of *T*_g_ was evidenced. This could be explained considering that not polymerizable ILs remained as liquid in the formulation, swelling the matrix and acting as plasticizers. On the other hand, when the ILs entered in the polymeric network, the chain mobility decreased, leading to an increase of *T*_g_. The results followed a similar trend both on DSC and DMA, confirming the assumptions. For DMTA, unfortunately, it was not possible to perform measurements on P_Amim[Tf_2_N] sample due to brittleness and poor mechanical properties of this membrane.

According to literature [56], DMTA measurements were used also to calculate the cross-linking density according to the Equation
*E*′ = υ·*R*·*T*(1)
where *E*′ (MPa) is the conservative modulus in the rubbery plateau, υ (mol/cm^3^) is the cross-linking density, *R* (cm^3^ MPa/mol K) the gas constant and *T* the temperature in *K* at which the measure is performed (in our case, 293 K for all the samples). The values (reported in Table 3) confirmed that the ILs without reactive groups drastically reduced the cross-linking density, acting as swelling agent in the polymeric matrix. A different behavior was measured for the IL with a single cross-linkable moiety (C_4_N_MA,11_[Tf_2_N]): even in this case the cross-linking density decreased as expected, since a monofunctional monomer was introduced in the network. On the other hand, the *T*_g_ slightly increased, related to the fact that the IL was not anymore liquid but integrated in the network. This was measured even in the presence of Bvim[Tf_2_N]. Moreover, the sample P_Bvim[Tf_2_N] showed also an increase of cross-linking density, related to the fact that this IL is bifunctional, thus introducing additional cross-linking points.

The first approach on the study of the interaction between CO_2_ and the polymerized structures containing dispersed and linked ionic liquid species was to carry out gas barrier measurements on the membranes. These experiments are fundamental for the evaluation of the gas transport properties such as permeability, diffusivity and solubility. Furthermore, they are a first step towards the understanding of the usage of these materials as supported liquid membranes (SLM) for CO_2_ separation. The membranes were tested with the same IL concentrations (0.4 mmol/g) in order to evaluate the real effect of each IL. The values of diffusivity, solubility and permeability, calculated from the CO_2_ transmission rate curves of the membranes as described in a precedent work [57], are shown in Table 4. With the addition of ionic liquids in the formulations, the diffusivity increases in all cases, with the exception of C_4_N_MA,11_[Tf_2_N]. The best result is the one with Bmim[BF_4_] that presents a diffusivity value almost double with respect the neat PEGDA. For what regards solubility, the only sample with a significant variation from neat PEGDA is Amim[Tf_2_N] that unfortunately has got issues in the 3D printing process. The characteristic that allows to evaluate the effectiveness of a SLM system is the permeability and, among the analyzed samples, the most promising are the Bmim[BF_4_] and Bmim[Tf_2_N].

Afterwards, further permeability tests were conducted by increasing the amount of Bmim[BF_4_] and C_4_N_MA,11_[Tf_2_N] in the printable formulations. The change in the amount of the two ILs did not affect the printability of the resins and the same irradiation times were used for producing the membranes to be tested. Those two ILs were selected among the others because they seemed to give the best (Bmim[BF_4_]) and the worst C_4_N_MA,11_[Tf_2_N] performance when introduced in PEGDA. In Appendix A of the supporting information the values of diffusivity, solubility and permeability are displayed with the increase of the ILS concentration. Unfortunately, it is not present a clear trend between the amount of the ILs and the gas permeation properties. Surprisingly, increasing the amount of C_4_N_MA,11_[Tf_2_N] a great increase of both solubility and diffusivity was measured, indicating that an optimization work could be performed individuating the most effective amount of IL. This is part of ongoing studies.

The interaction between polymers containing ILs and carbon dioxide was monitored also by FTIR. As an example, in Figure 3, the absorption spectra of P_Bmim[BF_4_] in the range between 2300 and 2400 cm^−1^ after different steps was reported. While after the production step the two typical peaks of this gas, related to the IR active ν_3_ vibrating mode centered at 2330 and 2360 cm^−1^ [58,59], were not present in the sample, as expected, they appeared after the treatment in CO_2_ atmosphere. The following treatment in vacuum produced a reduction of the absorbance in the range, indicating desorbing of CO_2_. However, CO_2_ was still present in the film due to the interaction with the embedded IL. A following thermal step led to a further but incomplete desorption of CO_2_, indicating that IL-gas interaction is strong and could be used for gas trapping.

At last, taking advantage of the high printability of the developed formulations, we printed components of very complex geometry like gyroids. In Figure 4 we reported gyroids of different sizes printed with P_Bmim[BF_4_] formulation, but the same structures were obtained with the all the other tested ILs except P_Amim[Tf_2_N]. This highly complex geometry applied to 3D printing offers improved mechanical properties compared to other structures [60] and a high surface to volume ratio [61] that can be exploited to create active filter for CO_2_ capture with improved gas/surface interaction. In particular, these shapes present interconnected continuous channels that allow the passage of the gas without a pressure drop but, at the same time, higher interaction with the structure for enabling gas adsorption. Those complex shape components are at the moment under investigation as well and will be presented in future works. In fact, different parameters should be taken into account in order to have an optimized device including material properties, channel dimensions, tortuosity and available surface. In this context 3D printing could be fully exploited integrating CAD design with formulation development.

## 4. Conclusions

In this work, we tested different commercially available ionic liquids in order to develop 3D printable formulations processable with DLP technology, with potential application in CO_2_ capture technology. Six ionic liquids were tested (Bmim[BF_4_], Bmim[Tf_2_N], Bvim[Tf_2_N], Amim[Tf_2_N], Bmim[ac] and C_4_N_MA,11_[Tf_2_N]) mixed with a bi-functional acrylate monomer (PEGDA). The photo-rheological analyses carried out brought us to exclude Bmim[ac] since the acetate anion inhibits the photopolymerization. Regarding the other ILs, we evidenced that imidazole based cations as well as tetrafluoroborate and bis (trifluorometilsulfonyl)imide anions did not affect the photopolymerization, while the other cations (containing ammonium salt and allyl group) slowed down the process. Afterwards, a 3D printing test was conducted on the formulations to build simple (membranes) and complex (hollow cubic) structures and to find the optimal printing parameters. All the ILs samples allowed the fabrication of complex 3D structures, with the exception of the samples containing Amim[Tf_2_N]. Moreover we demonstrated that the ILs bearing a reactive group (C_4_N_MA,11_[Tf_2_N] and Bvim[Tf_2_N]) participated to the polymerization reaction entering in the polymeric network, while it was possible to remove the other ILs by solvent extraction. This resulted even in thermomechanical properties, with a plasticizing effect of the dispersed ILs. The samples were then tested for the CO_2_ capture capacity by means of FTIR. The spectra showed the characteristic peaks related to the vibrating modes of the CO_2_ molecule, demonstrating the physical adsorption of the gas in the samples. Moreover an absorption/desorption experiment was conducted to check the regeneration of the samples. Regarding the possible application of these materials as supported liquid membranes (SLM) for CO_2_ separation, gas transport analyses were carried out to determine the transport properties of the membranes, unfortunately resulting in not univocal trends. As a final experiment, we 3D printed part with high complexity as gyroids in order to demonstrate the possibility to produce complex geometries that could be employed as active filters for CO_2_ capture. The present work successfully demonstrated the possibility to employ 3D printing for CO_2_ capture technologies, opening the way to new investigations in which CAD design and materials properties could lead to more efficient devices.

## Figures and Tables

**Figure 1 polymers-11-01932-f001:**
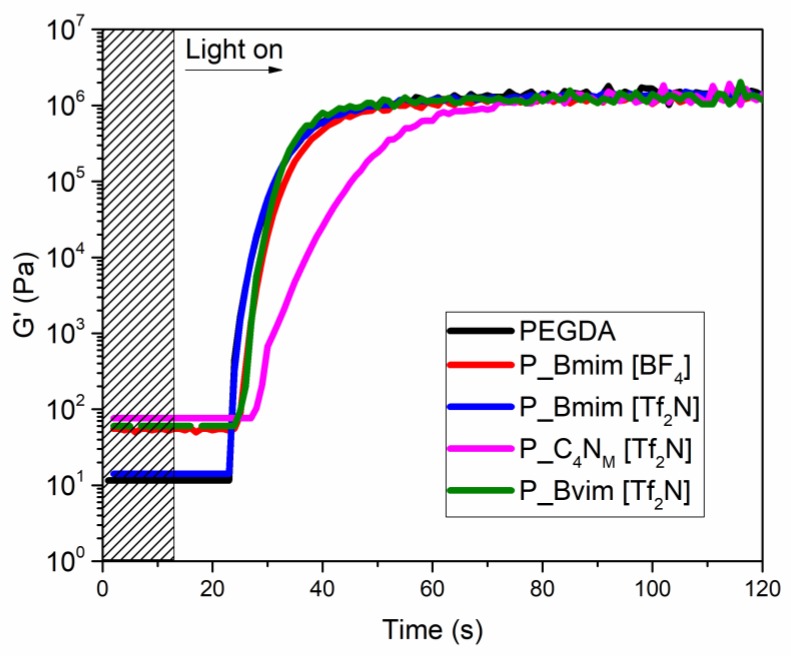
Photorheology tests of PEGDA/ionic liquid (IL) formulations containing different ILs and neat PEGDA as reference. Light irradiation starts after 12 s.

**Figure 2 polymers-11-01932-f002:**
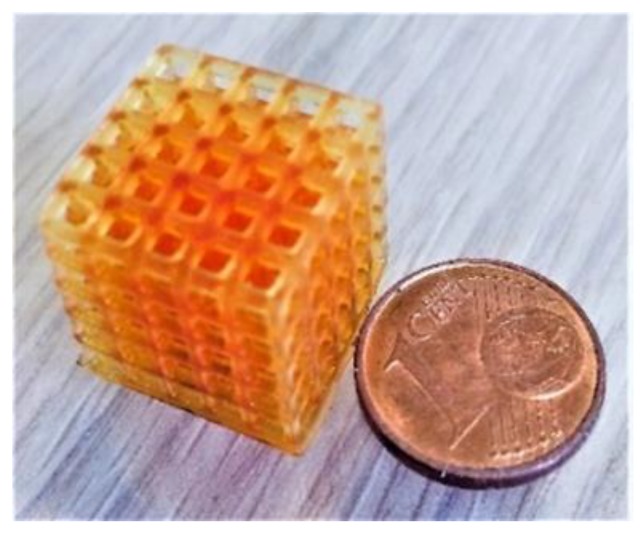
3D printed complex structure (hollow cube).

**Figure 3 polymers-11-01932-f003:**
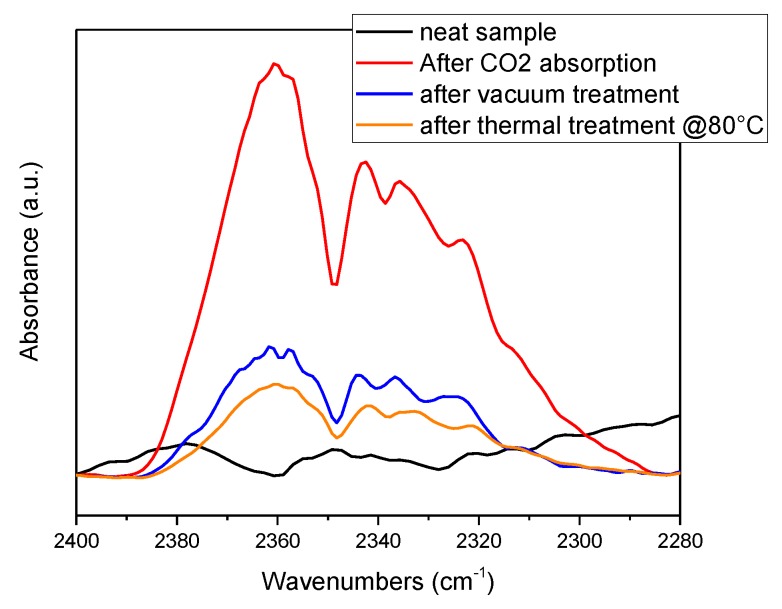
FT-IR of P_Bmim[BF4] in the CO_2_ absorption area after different absorption/desorption steps.

**Figure 4 polymers-11-01932-f004:**
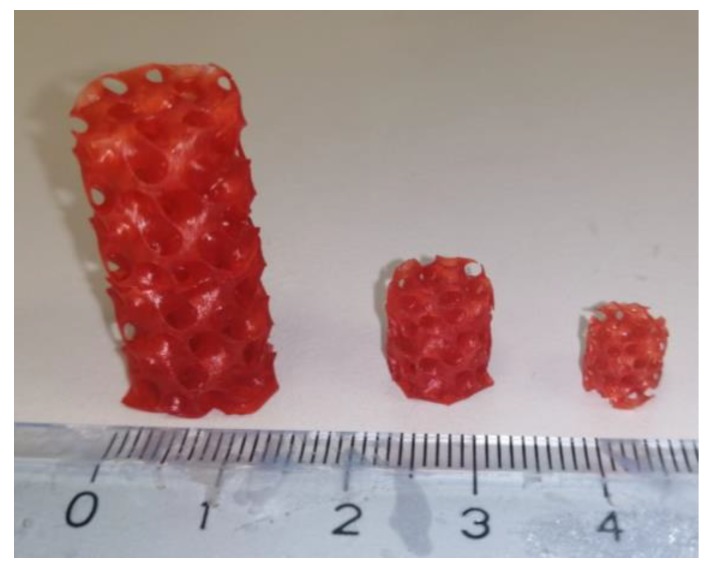
Gyroid-like structures printed with P_Bmim[BF_4_].

**Table 1 polymers-11-01932-t001:** Ionic liquids tested in this study.

Butyl-3-Methylimidazolium tetrafluoroborate Bmim[BF_4_] 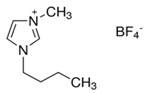	*N,N,N,N*-ButyldimethylMethacryloyloxyethyl ammonium bis(trifluoromethylsulfonyl)imideC_4_N_MA,11_[Tf_2_N] 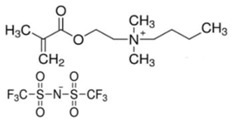
1-Butyl-3-Methylimidazolium bis (trifluorometilsulfonyl)imide Bmim[Tf_2_N] 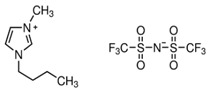	1-Allyl-3-Methylimidazolium bis (trifluoromethylsulfonyl)imide Amim[Tf_2_N] 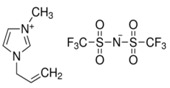
1-Butyl-3-Methylimidazolium acetate Bmim[ac] 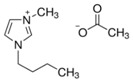	1,4-Butandiyl-3,3′-bis-1-vinylimidazoliumbis (trifluoromethylsulfonyl)imide Bvim[Tf_2_N] 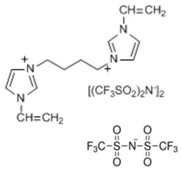

**Table 2 polymers-11-01932-t002:** Layer exposure times needed for the samples.

Sample	Layer Exposure Time (s)	
	Membrane	Cubic Filter-Like
PEGDA	1	2
P_Bmim[BF4]	1	1.7
P_Bmim[Tf2N]	1	2
P_C_4_N_MA,11_[Tf2N]	1	1.8
P_Amim[Tf2N]	2	Not obtained
P_Bvim[Tf2N]	1	1.7

**Table 3 polymers-11-01932-t003:** Ionic liquid content in the samples in relation to the weight variation after solvent treatment.

Sample	Ionic Liquid Concentration (wt %)	Weight Variation (wt %)	*T*_g_ (DSC; °C)	*T*_g_ (DMTA; °C)	υ (mmol/cm^3^)
PEGDA	0	0	−19.9	−17	10.8
P_Bmim[BF_4_]	9	8.5	−22.6	−21.3	1.3
P_Bmim[Tf_2_N]	15.1	13.9	−22.5	−21.6	1.4
P_C_4_N_MA,11_[Tf_2_N]	17.5	0	−15.1	−12.8	1.6
P_Amim[Tf_2_N]	14.7	14.2	−44.8	Not calculated	-
P_Bvim[Tf_2_N]	14.7	0	−11.2	−10	12.2

**Table 4 polymers-11-01932-t004:** Calculated diffusivity (D), solubility (S) and permeability (P) values for 3D printed membranes containing different ionic liquids.

Sample	$D\;\left(\frac{cm^{2}}{s}\right)$	S	$P\;\left(\frac{cm^{2}}{s\times bar}\right)$
PEGDA	6.73	4.71	3.17
P_Bmim[BF_4_]	11.33	2.99	3.39
P_Bmim[Tf_2_N]	8.65	5.26	4.55
P_C_4_N_MA,11_[Tf_2_N]	4.46	4.22	1.88
P_Amim[Tf_2_N]	9.53	14.47	13.79
P_Bvim[Tf_2_N]	8.45	3.20	2.70

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
