# Peer review of "Study on the Printability through Digital Light Processing Technique of Ionic Liquids for CO_2_ Capture"

_polymers, 2019, doi:10.3390/polym11121932_

Round 1

Reviewer 1 Report

The authors presented 3D printable materials based on the introduction of different commercially available Ionic Liquids (ILs) in the starting formulations. They evaluated the influence of these additives on the printability of such formulations through light-induced 3D printing (Digital Light Processing), investigating as well the effect of Ionic Liquids with polymerizable groups. The physical chemical properties of the materials were compared. The review comments for this paper are in the following

Please avoid using abbreviations in the TITLE, HIGHLIGHTS, and Keywords if possible such as DLP, etc.  

Line 30, "different technologies are available for CO2 capture and storage (CCS), such as pre-combustion, post-combustion and oxyfuel capture" -- The methods of pre-combustion, post-combustion and oxyfuel capture are only for CO2 capture not for CO2 storage. "The most of these technologies employ liquids" -- Absorption is mainly for the post-combustion method.

Please add the comparisons between ionic liquids with other CO2 solvents.

Regarding the 3D printing, what are the developments in the polymeric materials?

Line 80, please delete "The introduction should briefly place the study...the document for further details on references."

In Section 2.1, please provide the detailed information of the materials and chemicals. 

In Table 1, please improve the chemical structure of C4NMA,11[Tf2N].

Line 110, space "intensity" and "10". 

Please use the same format of "mWcm−2" and "mW/cm2".

Line 136, change "were" to "was".

Please extend the title of Figure 1. More detailed analysis of the results for various ILs should be added.

In Figure 2, why did you choose the cubical structure?

In Eq. (1), please give the units of these parameters. 

Line 198, change "Where" to "where".

In Table 3, change "%wt" to "wt%".

How could you calculate the diffusivity, solubility and permeability? In Figure S5, there is no units of these parameters.

Please double check the subscripts of CO2 in the whole text.

In Figure 4, please describe the important peaks of the FTIR results.

In Figure 5, the authors presented the gyroids of different sizes printed with P_Bmim[BF4] formulation. What are the potential for the CO2 adsorption capacity? What are the main controlling factors of the structures?

The authors need to add the references to verify the findings in this work. Please check the references carefully. The conclusions need to be improved. 

What are the stability of the 3D materials for CO2 sorption?

Author Response

On behalf of all the authors, I would like to thank the reviewers for their precious comments. WE have modified the manuscript according to their suggestions. Here we enclose a point by point reply to their doubts, highlighting our response in yellow.

We do hope that we were able to meet their points.

All the best

Ignazio Roppolo, Ph.D.

Reviewer 1

The authors presented 3D printable materials based on the introduction of different commercially available Ionic Liquids (ILs) in the starting formulations. They evaluated the influence of these additives on the printability of such formulations through light-induced 3D printing (Digital Light Processing), investigating as well the effect of Ionic Liquids with polymerizable groups. The physical chemical properties of the materials were compared. The review comments for this paper are in the following

Please avoid using abbreviations in the TITLE, HIGHLIGHTS, and Keywords if possible such as DLP, etc.

Revision: The title was modified accordingly, as well as the keywords.

Line 30, "different technologies are available for CO2 capture and storage (CCS), such as pre-combustion, post-combustion and oxyfuel capture" -- The methods of pre-combustion, post-combustion and oxyfuel capture are only for CO2 capture not for CO2 storage. "The most of these technologies employ liquids" -- Absorption is mainly for the post-combustion method.

Revision: The paragraph was modified addressing to carbon capture processes only and a more detailed distinction between adsorption and absorption has been provided.

Please add the comparisons between ionic liquids with other CO2 solvents.

Revision: a comparison between ionic liquids and aqueous alkanolamine, which are the most common solvents for CO2 capture, has been added.

Regarding the 3D printing, what are the developments in the polymeric materials?

Revision: this part was updated introducing corresponding literature.

Line 80, please delete "The introduction should briefly place the study...the document for further details on references."

Revision: We are sorry for forgetting this part during the copy/paste of the text in the journal format. This part was deleted.

In Section 2.1, please provide the detailed information of the materials and chemicals.

Revision: The section was update with information of the chemicals employed.

In Table 1, please improve the chemical structure of C4NMA,11[Tf2N].

Revision: The chemical structure was improved.

Line 110, space "intensity" and "10".

Revision: Thank you, the space was inserted.

Please use the same format of "mWcm−2" and "mW/cm2".

Revision: Thank you, the format was changed.

Line 136, change "were" to "was".

Revision: So sorry for the mistake, the verb was changed.

Please extend the title of Figure 1. More detailed analysis of the results for various ILs should be added.

Revision: The title was extended; Moreover we better described the different behaviors of the ILS, mentioning the effect of cation/anion on the photopolymerization process.

In Figure 2, why did you choose the cubical structure?

Revision: Thank you for the question, an explanation was added in the text. Due to our experience in 3D printing, hollow-cube structures are for us better benchmarks than standard structures proposed ( holes and pillars of different dimensions) for testing the possibility of printing complex structures. In particular, the presence of simple vertical pillars and a complex internal structure allow to check the construction along z axis and to remove properly unreacted resin trapped in the structure.

In Eq. (1), please give the units of these parameters.

Revision: the units were added.

Line 198, change "Where" to "where".

Revision: modified.

In Table 3, change "%wt" to "wt%".

Revision: changed.

How could you calculate the diffusivity, solubility and permeability? In Figure S5, there is no units of these parameters.

Revision: It has been added a reference (54) in which the methodology used is described. The units in the figures have been added.

Please double check the subscripts of CO2 in the whole text.

Revision: Sorry for forgetting, all checked.

In Figure 4, please describe the important peaks of the FTIR results.

Revision: the peaks have been described as vibrational modes of CO2 molecules centered at 2330 cm-1 and 2360 cm-1.

In Figure 5, the authors presented the gyroids of different sizes printed with P_Bmim[BF4] formulation. What are the potential for the CO2 adsorption capacity? What are the main controlling factors of the structures?

Revision: Gyroids are complex shape with continuous interconnected channels. The use of this shape could allow the flow of the gas increasing the volume/surface contact, so enabling a higher gas absorption without pressure drop (i.e. in case of membranes). Moreover they enable an optimized saving of materials. At last they could host other sorbents (i.e. powders) in the interconnected porosity, allowing an easy handling of an absorbing device. This part was revised, integrating the parameters on which we are currently working for the fabrication of 3D printed devices. 

The authors need to add the references to verify the findings in this work. Please check the references carefully. The conclusions need to be improved.

Revision: We deeply revised the conclusions as well as the references. 

What are the stability of the 3D materials for CO2 sorption?

Revision: Regarding the mechanical stability, the sample are stable over years, without cracking or degradation. Instead, the interaction with environmental CO2 and pollutants implies to perform regeneration cycles before every test.

Reviewer 2 Report

There are several issues with the manuscript that should be addressed before further consideration for publication.

Some of the original text, assuming from the template is still in the manuscript submitted. Please revise accordingly. 

1. The authors used DLP to prove the "printability" of the materials, however, there is no detail description of the process. What is DLP? What are the key parameters that affect the fabricated samples?

Lee et al. (2018), 3D bioprinting processes: A perspective on classification and terminology, International Journal of Bioprinting 4 (2), 151 Kadry et al. (2019), Digital light processing (DLP) 3D-printing technology and photoreactive polymers in fabrication of modified-release tablets, European Journal of Pharmaceutical Sciences 135, 6-67

2. Why are these particular formulations of the materials chosen?

3. Why are lattices chosen to show printability, instead of standard benchmark parts? What is measured to determine printability?

Yap et al. (2017), Material jetting additive manufacturing: An experimental study using designed metrological benchmarks, Precision Engineering 50, 275-285

4. One of the formulation is not able to produce the gyroid shape. Why is this so?

Author Response

On behalf of all the authors, I would like to thank the reviewers for their precious comments. WE have modified the manuscript according to their suggestions. Here we enclose a point by point reply to their doubts, highlighting our response in yellow.

We do hope that we were able to meet their points.

All the best

Ignazio Roppolo, Ph.D.

Reviewer 2

There are several issues with the manuscript that should be addressed before further consideration for publication.

Some of the original text, assuming from the template is still in the manuscript submitted. Please revise accordingly.

Revision: We are sorry for forgetting this part during the copy/paste of the text in the journal format. This part was deleted.

The authors used DLP to prove the "printability" of the materials, however, there is no detail description of the process. What is DLP? What are the key parameters that affect the fabricated samples?

DLP technology was better explained in the introduction. Moreover a sentence describing the key parameter to control was introduced in the text.

Why are these particular formulations of the materials chosen?

Revision: the monomer PEGDA Mn:575 was chosen because it is widely applied in DLP 3D printing and because, due to its chemical structure, it is a good candidate for applications in CO2 capture and separation, mainly related to the high solubility of CO2 if compared to other polymers (Ref 56). Regarding the introduction of ILs, they were added with a starting concentration of 0.4 mmol/g, which corresponds more or less to the introduction of 10 wt% of ILs in PEGDA, with obvious variation related to the Mw. Regarding the chosen ILs, we wanted to investigate similar cations ( imidazole) and different anions as well as different cations with the same anion, introducing even photocurable moieties. We aimed to have a first indication of what pair cation/anion was more promising for developing 3D printed devices.

Why are lattices chosen to show printability, instead of standard benchmark parts? What is measured to determine printability?

Revision: Thank you for the question, an explanation was added in the text. Due to our experience in 3D printing, hollow-cube structures are for us better benchmarks than standard structures proposed ( holes and pillars of different dimensions) for testing the possibility of printing complex structures. In particular, the presence of simple vertical pillars and a complex internal structure allow to check the construction along z axis and to remove properly unreacted resin trapped in the structure.

One of the formulation is not able to produce the gyroid shape. Why is this so?

Revision: Thank you for the question. Now we have highlighted in manuscript that Amim[Tf2N] gave polymerization issues related to allyl groups, that can be the responsible of the poor reactivity of the formulation. This led as well to have very poor mechanical properties, which resulted in very fragile and loose membranes. So the problem was that the polymerized structures were not able to self-sustain, disabling the growth of a 3D structure along z direction.

Round 2

Reviewer 1 Report

The paper title in the submission system is still the same. Where is Figure 3 in the main text?

Author Response

Authors appreciate for your comments and consideration.

We have contacted editors and updated the title in the manuscript. And we have revised the order of the figures in the manuscript.

All the best

Ignazio Roppolo, Ph.D.

Reviewer 2 Report

NA

Author Response

Many thanks for your comments and consideration.

All the best

Ignazio Roppolo, Ph.D.